# Comparison In Vitro Study on the Interface between Skin and Bone Cell Cultures and Microporous Titanium Samples Manufactured with 3D Printing Technology Versus Sintered Samples

**DOI:** 10.3390/nano14181484

**Published:** 2024-09-12

**Authors:** Maxim Shevtsov, Emil Pitkin, Stephanie E. Combs, Greg Van Der Meulen, Chris Preucil, Mark Pitkin

**Affiliations:** 1Department of Radiation Oncology, Technische Universität München (TUM), Klinikum Rechts der Isar, Ismaninger Str. 22, 81675 Munich, Germany; stephanie.combs@tum.de; 2Laboratory of Biomedical Nanotechnologies, Institute of Cytology of the Russian Academy of Sciences (RAS), 194064 Saint Petersburg, Russia; 3Personalized Medicine Centre, Almazov National Medical Research Centre, 2 Akkuratova Str., 197341 Saint Petersburg, Russia; 4Department of Statistics and Data Science, The Wharton School, University of Pennsylvania, Philadelphia, PA 19104, USA; emil.pitkin@gmail.com; 5Movora, St. Augustine, FL 32095, USA; greg.vandermeulen@movora.com (G.V.D.M.); chris.preucil@movora.com (C.P.); 6Department of Orthopaedics and Rehabilitation Medicine, Tufts University School of Medicine, Boston, MA 02111, USA; 7Poly-Orth International, Sharon, MA 02067, USA

**Keywords:** osteoblasts, fibroblasts, osseointegration, 3D printing, titanium alloy, bone tissue engineering, scaffolds

## Abstract

Percutaneous implants osseointegrated into the residuum of a person with limb amputation need to provide mechanical stability and protection against infections. Although significant progress has been made in the biointegration of percutaneous implants, the problem of forming a reliable natural barrier at the level of the surface of the implant and the skin and bone tissues remains unresolved. The use of a microporous implant structure incorporated into the Skin and Bone Integrated Pylon (SBIP) should address the issue by allowing soft and bone tissues to grow directly into the implant structure itself, which, in turn, should form a reliable barrier to infections and support strong osseointegration. To evaluate biological interactions between dermal fibroblasts and MC3T3-E1 osteoblasts in vitro, small titanium discs (with varying pore sizes and volume fractions to achieve deep porosity) were fabricated via 3D printing and sintering. The cell viability MTT assay demonstrated low cytotoxicity for cells co-cultured in the pores of the 3D-printed and sintered Ti samples during the 14-day follow-up period. A subsequent Quantitative Real-Time Polymerase Chain Reaction (RT-PCR) analysis of the relative gene expression of biomarkers that are associated with cell adhesion (α2, α5, αV, and β1 integrins) and extracellular matrix components (fibronectin, vitronectin, type I collagen) demonstrated that micropore sizes ranging from 200 to 500 µm of the 3D printed and sintered Ti discs were favorable for dermal fibroblast adhesion. For example, for representative 3D-printed Ti sample S6 at 72 h the values were 4.71 ± 0.08 (α2 integrin), 4.96 ± 0.08 (α5 integrin), 4.71 ± 0.08 (αV integrin), and 1.87 ± 0.12 (β1 integrin). In contrast, Ti discs with pore sizes ranging from 400 to 800 µm demonstrated the best results (in terms of marker expression related to osteogenic differentiation, including osteopontin, osteonectin, osteocalcin, TGF-β1, and SMAD4) for MC3T3-E1 cells. For example, for the representative 3D sample S4 on day 14, the marker levels were 11.19 ± 0.77 (osteopontin), 7.15 ± 0.29 (osteonectin), and 6.08 ± 0.12 (osteocalcin), while for sintered samples the levels of markers constituted 5.85 ± 0.4 (osteopontin), 4.45 ± 0.36 (osteonectin), and 4.46 ± 0.3 (osteocalcin). In conclusion, the data obtained show the high biointegrative properties of porous titanium structures, while the ability to implement several pore options in one structure using 3D printing makes it possible to create personalized implants for the best one-time integration with both skin and bone tissues.

## 1. Introduction

Several research groups have attempted to reduce the infection rate post-implantation of percutaneous implants [1,2,3,4,5]. For example, in the Jeyapalina et al. study, the authors showed how the skin infection rate was reduced to 16.7% over a 24-month period in an ovine amputation model [6]. However, 25% of the sheep were removed due to early complications. Also, despite initial skin ingrowth into the pylon, a down-growth of skin epithelium along the implant broke the skin seal and introduced the potential risk of future infection and implant failure. This negative result may be explained in part by the fact that the researchers either did not use porous titanium at all in the implant design or employed relatively thin porous cladding, which in both cases did not lead to the desired biointegration of the implant with the surrounding tissues [7,8,9].

In contrast, the Poly-Orth International team has created a novel biotechnological platform called the Skin and Bone Integrated Pylon (SBIP) [10,11]. The SBIP is a patented, deeply porous transcutaneous implant that uses the natural anisotropy of skin regeneration to help establish functional safety in the skin-device interface. Numerically, the depth of porosity can be calculated with a parameter called “volume fraction”, which is defined as the ratio of the volume of the porous portion to the entire volume of the device. The success of small and large animal studies with SBIPs [12,13,14,15,16,17,18,19,20,21,22,23,24,25,26,27,28,29,30,31,32,33,34,35,36,37] can be attributed to the fact that the porosity of the samples was not superficial, as before, but deep (>50%).

For previous in vitro and in vivo preclinical studies SBIP samples were fabricated by sintering titanium particles in boron nitride molds [10,11,12]. That technology has some intrinsic inadequacies that render it unacceptable for translation to human applications. It does not allow for exact prediction of the pore sizes and cannot provide sufficient strength to the porous cladding.

A promising alternative to sintering technology in fabricating implants for osseointegration is additive manufacturing (3D printing), which is rapidly developing and is actively expanding into the field of osseointegration [5,38,39,40,41,42]. In addition to the greater strength and rapid fabrication of individual devices, specifically for percutaneous osseointegrated implants, 3D printing may support a different structure of the porous cladding for its interface with the bone canal and the surrounding skin. Verification of the latter is the main goal of the current study.

For the assessment of the Ti scaffold biocompatibility, we employed dermal fibroblasts and MC3T3-E1 osteoblast cells. The interaction of cells with the surface of implants comprises several stages, including cell adhesion and spreading, clustering of receptors in focal adhesions, production of stress fibers, and subsequent proliferation and differentiation on the material surface [43,44,45,46]. Indeed, these processes could be used to estimate implant biocompatibility [43]. Accordingly, the processes of cell adhesion as well as the production of extracellular matrix (ECM) can be monitored by quantitative and qualitative analysis of the various proteins produced by cells. For fibroblasts, we estimated the expression of α2 integrin (collagen-specific), α5 integrin (fibronectin-specific), αV integrin (vitronectin-specific), and β1 integrin genes, as well as expression of collagen, vitronectin and fibronectin genes. For MC3T3-E1 cells, we assessed focal adhesion markers, including FAK, vinculin, and paxillin. Apart from these molecules, we additionally assessed specific genes involved into osteogenesis (i.e., osteopontin, osteonectin, and osteocalcin) as well TGF-β1 and SMAD4 involved in the signaling pathway regulating this process. Indeed, as was shown previously, growth factor TGF-β1 (via canonical and non-canonical pathways) plays a key role in osteoblast growth and differentiation and the regulation of osteoclastogenesis [47,48,49]. A major transducer of BMP (bone morphogenetic protein) and TGF-β signaling pathways, SMAD4, regulates osteoblast and osteocyte viability [50].

In the current study, we aimed to demonstrate that 3D printed Ti discs with defined micropore size varying for different skin and bone cell types have a good biocompatibility profile that facilitates fibroblast and osteoblast cell attachment, ingrowth, proliferation, and osteogenic differentiation. The data presented supports the application of microporous titanium implants manufactured by 3D printing technology in regenerative medicine.

## 2. Materials and Methods

### 2.1. Samples for the Current Study

Experimental samples were 3D printed from medical-grade titanium. Our previous studies showed that skin and bone cells exhibit better interactions with irregular pores compared to those with regular pores [51,52]. Therefore, we developed the 3D printing process to have a mix of pores within a certain range of sizes. The samples were shaped as cylindrical tablets with a thickness of 10 mm and an outer diameter of 13.6 mm. The tablets have a solid core (D = 6.8 mm) surrounded by porous cladding (Figure 1).

Nine sets (S1–S9) of 12 tablets each were fabricated with average pore sizes ranging from 210–1000 µm (Table 1).

**Table 1 nanomaterials-14-01484-t001:** Lattice structure design parameters of the 3D printed Ti samples. The strut diameters and pore sizes were taken as an average of 5 measurements obtained with a Dino-Lite Camera at a magnification of 50.6. Sample Identifier refers to the number that was engraved on each different sample tab. %Solid represents the percentage of the total volume of the lattice structure that is metal. %Porous represents the percentage of the total volume of the lattice section that is void of material. Strut Diameter refers to the diameter of the lattice beams. Average pore diameter represents the average diameter of a sphere that can lie tangent to the surrounding lattice beams (see Figure 2).

Sample Identifier	% Solid	% Porous	Strut Diameter (µm)	Average Pore Diameter (µm)
S1	28.6	71.4	270	1000
S2	32.7	67.3	270	890
S3	37.3	62.7	270	770
S4	43.2	56.8	260	690
S5	50.0	50.0	270	590
S6	58.7	41.3	270	500
S7	78.8	21.2	250	420
S8	68.9	31.1	250	310
S9	80.5	19.5	230	210

Control samples were tablets (Figure 3) fabricated with sintering technology with pore sizes that had demonstrated better results in our previous studies. The size of compacted and sintered particles was within the range of 500 µm.

We used a patented combination of four key technological characteristics: porosity, pore size, porosity volume fraction, and particle size [10,11]. The parameter most distinct from other implants’ systems is the porosity volume fraction (VF).

For the cylindrical devices, volume fraction can be calculated with Formula (1),
(1)$$VR={(r}_{1}^{2}-r_{2}^{2})/r_{1}^{2}$$
where *r*_1_ is the outer radius of the tablet and *r*_2_ is the radius of the core. In our samples, *VF* = 78.2%. This value is associated with implants with deep porosity (*VF* > 50%) as defined in [12,31].

### 2.2. Specifics of 3D Printing Fabrication

The tablets were initially modeled in Solidworks (Dassault Systèmes, GSC, Germantown, WI, USA) to define the solid and porous bodies of each tablet. The models were next imported into 3-Matic (Materialise, Leuven, Belgium) as an assembly, and the varying and porous bodies with the aforementioned varying lattices were created. The models of the completed tablets were imported into Magics (Materialise, Leuven, Belgium) to create a build layout and slice the files for 3D printing (Figure 4a) This file was exported to an M2 Series 5 printer (Colibrium Additive, Rock Hill, SC, USA), and the tablets were printed. They were removed from the build plate (Figure 4b) using a standard Wire EDM (Wire Discharge Machining, Waukesha, WI, USA). Each of the tablets was then bead blasted (40–60 psi, glass bead—150–212 Micron) to remove any unsolidified powder and ultrasonically cleaned.

### 2.3. Cell Culture

Human dermal fibroblasts were cultivated in DMEM supplemented with 10% fetal bovine serum (FBS), 6  mM L-glutamine, 1  mM MEM Sodium Pyruvate, 0.1  mM MEM Non-Essential Amino Acids 4.5  g/L Glucose, and 1% Pen/Strep (Gibco, Waltham, MA, USA). Cells were passaged at least three times prior to co-incubation with Ti implants. Mouse pre-osteoblastic MC3T3-E1 (ATCC, CRL-2594) cells were cultured in α-minimum essential medium (α-MEM) (Gibco, Waltham, MA, USA) supplemented with 10% (FBS) (Gibco, Waltham, MA, USA) and 1% penicillin-streptomycin (Gibco, Waltham, MA, USA) at a 37 °C and 5% CO_2_. 0.25% trypsin-EDTA (Gibco, Waltham, MA, USA) was employed for cell dissociation at high cell confluence (≥90%). To assess the osteoinductive activity of 3D-printed Ti samples, MC3T3-E1 cells were cultured in an osteoinductive α-MEM medium, which contained 10 mM β-glycerophosphate (Sigma-Aldrich, St. Louis, MO, USA), 50 μg/mL of l-ascorbic acid (Sigma-Aldrich, St. Louis, MO, USA), and 100 nM of dexamethasone (Sigma-Aldrich, St. Louis, MO, USA).

### 2.4. Cell Proliferation Assay

Dermal fibroblasts and MC3T3-E1 cells proliferation was measured by 3-[4,5-dimethylthiazol]-2, 5-diphenylterazolium bromide assay (MTT assay) (Invitrogen, Waltham, MA, USA). In brief, cells were plated at a density of 5 × 10^3^ cells/well in 96-well plates. After seeding Ti implants with cells, the samples were co-incubated for 1, 3, 7, and 14 days at 37 °C, 5% CO_2_. Following co-incubation 20 μL MTT (0.5 mg/mL) was added to each well and incubated at 37 °C for 4 h. The plate was read at 490 nm using a microplate reader (Bio-Rad, Hercules, CA, USA, model 550). All tests were performed in triplication independently. Proliferation cell rate (%) = (sample OD − blank OD)/(control OD − blank OD) × 100%

### 2.5. Quantitative Real-Time PCR

Total RNA from the dermal fibroblasts and MC3T3-E1 cells on each material (sintered and 3D-printed Ti) from all the assessed discs at each time point was extracted with the Qiagen RNA Plus kit (QIAGEN, Venlo, The Netherlands). The isolated total RNA was quantified using a nanodrop spectrophotometer (Thermo Scientific, Waltham, MA, USA). cDNA synthesis was performed for the isolated RNA samples and used for real-time PCR experiments. In brief, cDNA synthesis was performed using the superscript III reverse transcriptase (RT) enzyme (Invitrogen, Waltham, MA, USA). 2 μg RNA was added to a reaction mix (10 mM deoxyribonucleotide triphosphate mix (dNTP), 50 μM oligodT) for first strand synthesis (65 °C, 5 min) with subsequent cooling on ice for 2–3 min. Then a mix containing the superscript III reverse transcriptase, RNase inhibitor, 0.1 M DTT (Di-thio-threitol), 5× reaction buffer was added to the first strand synthesized mixture, incubated for 1 h at 50 °C with a subsequent inactivation of RT for 15 min at 70 °C. cDNA for GAPDH was used as a control for calculating fold differences in RNA levels of fibroblasts and osteoblasts cultured on Ti discs. Forward and reverse primers specific for tested genes were designed with Pubmed nucleotide design (Primer-BLAST) software version 1.0.1 for all tested genes (Table 2). The samples were evaluated in the Applied Biosystems 7900HT Fast real-time PCR system (Applied Biosystems, Waltham, MA, USA) according to the manufacturer’s protocol.

### 2.6. Statistics

Each biomarker and extracellular matrix component was measured on each of nine experimental discs and one control disc for either four or two time periods, depending on the specific variable. Three samples were collected for each marker/experimental disc combination, and six samples for each variable/control disc combination. Means and standard deviations were computed in Excel. For each marker, bar charts were generated with bars representing the average measurement of each variable for each of the tablets with +/− 1 SD error bars overlayed. Because marker expression during the final time period is most clinically relevant, a one-way ANOVA, which tests for equality of means, was run for each marker for only the last time period, with tablet ID as the independent variable. Equal within-group variances were assumed, and alpha = 0.05 was the critical level for each test. Whenever the ANOVA showed significant differences in means, apost-hoc Tukey-Kramer test was performed for specific comparisons between individual tablets or groups of tablets.

## 3. Results

### 3.1. Evaluation of the Fibroblasts and MC3T3-E1 Viability on the Sintered and 3D Printed Ti Discs

The in vitro assessment of cell viability was performed with an MTT assay when cells (dermal fibroblasts, MC3T3-E1 osteoblasts) were co-cultured on sintered Ti samples (n = 9) for 1, 3, 7, and 14 days (Figure 5, Table 3 and Table 4). In the blank control group, when cells were cultured in cultural flasks without sintered or 3D-printed implants, the cell viability (%) for dermal fibroblasts on the 1st, 3rd, 7th, and 14th days were 97.41 ± 1.18%, 94.32 ± 2.21%, 86.78 ± 4.61%, and 85.53 ± 4.19%, respectively. The cell viability (%) in the blank control group for MC3T3-E1 osteoblasts on the 1st, 3rd, 7th, and 14th days was 95.37 ± 3.21%, 96.52 ± 1.89%, 89.91 ± 4.81%, and 84.92 ± 3.73%, respectively.

In the first three days of co-incubation, there was no significant decrease in cell viability in any of the studied samples (S1–S9) for either cell type (Figure 5, Table 3 and Table 4). Correspondingly, on day 3 the mean values for the fibroblasts ranged from 94.00 ± 3.48% to 96.87 ± 1.71%. Starting from day 7, there was a slight decrease in the viability of fibroblasts and osteoblasts, while no difference was detected between the samples. For fibroblasts, the cell viability ranged from 85.43 ± 2.43% to 89.67 ± 2.97%. For osteoblasts, the cell viability ranged from 84.93 ± 1.21% to 87.30 ± 1.71%. Next, we evaluated the cell viability culture on the sintered Ti discs (Figure 5, Table 3 and Table 4). Following co-incubation, we did not detect a significant influence on cell viability, although we observed a slight decrease of viability to around 80% on the 14th day of co-incubation with both types of cells. When cell viability was compared between 3D-printed samples (S1–S9) and sintered Ti samples, we did not observe any significant difference (*p* > 0.05), thus indicating good biocompatibility for both types of samples.

### 3.2. Analysis of Focal Adhesion Markers of Fibroblasts and Osteoblasts Cultured on Sintered and 3D Printed Ti Discs

The second stage of the work was to study adhesion molecules—integrins and related molecules during co-incubation of dermal fibroblasts with sintered and 3D-printed Ti samples (Figure 6, Table 5). Integrins play one of the most important roles in the interaction of cells with any substrate, including Ti. Gene expression of α2 integrin (collagen-specific), α5 integrin (fibronectin-specific), αV integrin (vitronectin-specific), and β1 integrin genes was assessed at 4, 24, 48, and 72 h of co-incubation on 3D-printed discs.

As can be seen in Figure 6 and Table 4, the expression of the studied markers increased starting from sample S1, reaching maximum values after 72 h for samples S6–S8, after which a decrease in expression was observed on sample S9. For example, for sample S6 at 72 h the values were 4.71 ± 0.08 (α2 integrin), 4.96 ± 0.08 (α5 integrin), 4.71 ± 0.08 (αV integrin), and 1.87 ± 0.12 (β1 integrin). It is worth noting that these same samples also showed high expression of collagen, vitronectin, and fibronectin. For example, for sample S6 at 72 h the values were 4.64 ± 0.57 (collagen), 4.64 ± 0.33 (vitronectin), 4.70 ± 0.09 (fibronectin). At the next stage, we assessed the gene expression of α2, α5, αV, and β1 integrin genes as well as collagen, vitronectin, and fibronectin at 4, 24, 48, and 72 h of co-incubation on sintered Ti samples (Figure 6; Table 5). As expected, we observed a gradual increase in the expression of the studied markers on the sintered Ti samples that reached their maximum on the 3rd day of co-incubation. The marker levels were comparable to the S6–S9 3D-printed Ti samples and were much higher than the S1–S5 3D-printed samples. For example, the mean values after 72 h of co-incubation for sintered Ti samples were 4.40 ± 0.14 (α2 integrin), 4.69 ± 0.16 (α5 integrin), 4.73 ± 0.32 (αV integrin), 1.89 ± 0.15 (β1 integrin), 4.42 ± 0.22 (collagen), 4.69 ± 0.10 (vitronectin), and 4.55 ± 0.11 (fibronectin).

The expression of focal adhesion markers, namely FAK, vinculin, and paxillin, on MC3T3-E1 osteoblast cells, was tested on the 3D printed materials and is presented in Figure 7, Table 6.

In all samples, a gradual increase in marker expression was observed, starting from the 3rd day, with samples S3–S5 showing the best results compared to other samples, reaching maximum values on the 14th day of observation. For example, for sample S4 on day 14, the values were 4.2 ± 0.09 (FAK), 4.1 ± 0.17 (vinculin), and 5.69 ± 0.37 (paxillin).

### 3.3. Analysis of Osteogenic Markers of MC3T3-E1 Cells Cultured on Sintered and 3D-Printed Ti Samples

In addition to the studied adhesion-related markers that are expressed by cells after attachment to substrates, the expression pattern of genes involved in osteogenesis on the surfaces of implant materials was examined at four time points—1, 3, 7, and 14 days after induction osteogenesis. The expression pattern of osteopontin, osteonectin, and osteocalcin, specific for osteogenesis of MC3T3-E1 cells co-cultured on sintered and 3D-printed Ti samples at different time points is shown in Figure 8 and Table 7.

As can be seen in the presented histograms and tables, the expression of the studied markers increased starting from sample S1, reaching maximum values for samples S3–S5, after which a decrease in expression was observed starting from S6 to reach the minimum levels for sample S9. Already starting from day 3 of co-incubation, a significant increase in the expression of these markers was observed. Thus, for 3D-printed sample S4 at day 3 of co-incubation, the values were 5.42 ± 0.54 (osteopontin), 3.92 ± 0.48 (osteonectin), and 3.29 ± 0.11 (osteocalcin). At the same time, it is worth noting that samples S3–S5 demonstrated the highest levels of expression of the studied markers on the 7th and 14th days. Thus, for representative sample S4 on day 14, marker levels were 11.19 ± 0.77 (osteopontin), 7.15 ± 0.29 (osteonectin), and 6.08 ± 0.12 (osteocalcin). At the same time on day 14, the levels for sintered samples were 5.85 ± 0.4 (osteopontin), 4.45 ± 0.36 (osteonectin), and 4.46 ± 0.3 (osteocalcin).

The expression of TGF-β1 and SMAD4, which play an important role in osteogenesis, was also assessed on MC3T3-E1 cells on days 1 and 7 (Figure 9, Table 8) for sintered and 3D-printed Ti discs.

The expression of TGF-β1 did not differ significantly in any studied samples after 7 days of co-incubation. Thus, for 3D-printed samples on day 7, the values ranged from 0.64 ± 0.06 to 0.84 ± 0.03. Intriguingly, for sintered Ti discs, we observed a significant increase in the levels of TGF-β1 of 1.91 ± 0.15.

SMAD4 expression increased in samples S1–S5 by day 7, with samples S3–S5 showing the best values. Thus, for 3D printed samples S3, S4, S5 on day 7 the level of SMAD4 was 1.75 ± 0.07 (S3), 1.79 ± 0.36 (S4), and 2.02 ± 0.16 (S5). The level of SMAD4 expression for sintered Ti was comparable to 3D-printed samples S6–S9 and was 1.17 ± 0.06.

## 4. Discussion

The development of new biocompatible nanomaterials to improve the efficiency of implant integration is one of the promising areas in modern translational and clinical traumatology and orthopedics [53,54,55]. In this study, for the first time, using 3D printing technologies within a single implant design, the possibility of selecting the optimal characteristics of the structure itself (pore size) for cells of both connective and bone tissues was presented. To date, using various materials, including Ti6Al4V, the size, and structure of scaffold pores have been shown to affect the process of osteogenesis, which includes cell adhesion and proliferation of osteoblasts and the formation of a mineralized bone matrix [56,57,58,59]. In our study, the 3D-printed samples S3–S5 with a pore size ranging from 400 to 800 μm showed the best results for MC3T3-E1 osteoblast cells (Figure 7, Figure 8 and Figure 9). In the current study, we observed a gradual increase in the expression of the marker levels starting from sample S1, reaching their maximum for samples S3–S5 with a subsequent decrease in expression from sample S6 to S9. The influence of pore size and structure on osteogenesis is still unresolved [60,61,62]. For example, Wang et al. had previously suggested that the optimal range for pore size of the implant ranges between 100–400 μm [61]. To the contrary, our data are in line with the reported results by Chen et al. who demonstrated that among selective laser melted (SLM) porous Ti6Al4V ELI scaffolds of 500 μm, 600 μm, and 700 μm the scaffold with porosity of 60% and pore sizes of 500 μm exhibited the best results in terms of osteogenic differentiation and bone ingrowth in the Sprague-Dawley rat model with the femoral condyles defect [63]. Furthermore, other studies also confirmed the preferable range of 400–600 μm for osteogenesis [42,64,65,66,67]. For example, Li et al. demonstrated that a Ti6Al4V scaffold with pore size 300–400 μm that was implanted into a large segmental defect of goat metatarsus resulted in bone ingrowth as compared to other pore size implants [42]. In another study, in rabbits with distal femoral defect, electron beam melting (EBM) fabricated porous titanium implants (383.2 μm pore size with 65.2% porosity; pores of 401.6 μm with 78.1% porosity) induced cell differentiation and bone ingrowth [65]. Ran et al. further showed that a porous scaffold with pores of 607 ± 24 µm (p700), produced with a 3D printing technique of selective laser melting (SLM), was favorable for bone-implant fixation stability [66].

In our study, 3D-printed samples differed only in the pore sizes; however, other parameters (e.g., surface roughness, the effect of the elastic modulus, etc.) might also contribute to the cell attachment and bone ingrowth [68,69,70,71]. Implant surface roughness can significantly affect the cell attachment, proliferation, and differentiation processes [9,72,73,74]. For example, Zemtsova et al. fabricated hierarchical nanotopographic (<100 nm)/microtopograhic (0.1–0.5 μm) coatings on Ti implants using the method of molecular layering of atomic layer deposition (ML-ALD) and proved the enhanced osteogenic differentiation of MC3T3-E1 osteoblast cells [73]. Subsequently, in a rabbit model of below-knee amputation, it was shown that these implants increased osseointegration strength (as measured by removal torque measurements) and enhanced bone formation in the zone of the bone-implant attachment [73]. Other studies further exploited this approach of generating a rough implant surface and producing hierarchical structures [75,76,77]. Parisien et al. evaluated the Ti6Al4V lattice structure’s geometrical parameters on the bone ingrowth stimulations that varied in pressure (0.5, 1, 1.5, and 2 MPa) and relative densities (ranging from 5 to 50%). It was shown in the subgroup of bending-dominated lattice topologies that the best topologies included BCC, FBCC, Diamond, Octahendron, and G7, while in the subgroup of stretching-dominated lattice topologies, Tesseract and Tetrahedron were best [68]. Thus, Wieding et al. implanted open-porous Ti6Al4V scaffolds (with different pore sizes and 6–8 GPa Young’s modulus) into the segmental defect (20 mm) of the sheep metatarsus and demonstrated new bone formation (analysis of bone mineral density, BMD) with good mechanical loading capabilities [78]. Subsequent studies further implemented this parameter into the design of the manufactured implants [79,80,81]. In our further preclinical studies, we also expect to implement these features (surface roughness, effect of the elastic modulus) in the produced implant.

Another cell type that was tested for biocompatibility of the 3D-printed Ti implants was dermal fibroblasts. Our data are in line with previously published reports that showed the absence of low cytotoxicity, cell attachment, and biological response [82,83,84]. According to our data, the preferable pore sizes for the attachment of dermal fibroblasts (expressions of integrins, collagen, vitronectin, and fibronectin) range from 200 to 500 μm (Samples S6–S8; Figure 6). In the current study, we observed a gradual increase in the expression of studied markers starting from sample S1, reaching a maximum for samples S6–S8 with a subsequent drop at sample S9. Indeed, as was described in a recent systematic review, porous titanium coatings with limited pore sizes (<250 μm) do not facilitate dermal fibroblast attachment [85]. Previously, Farrell et al. investigated skin-implant integration in rats when Ti porous implants (with pore sizes 40–100 μm (small) and 100–160 μm (large)) were percutaneously implanted. Following 6 weeks of observation, the authors reported skin ingrowth of over 50% of the total implant porous area under the skin, with a group of small pores showing a lower extrusion rate than the group with large pores implants (0.06 ± 0.01 vs. 0.16 ± 0.02 cm per week) [26]. In another in vivo study, porous Ti alloy cylinders (with a strut size of 300 μm and a pore size of 700 μm) implanted into sheep paraspinal muscles were favorable for soft-tissue ingrowth and revascularization [86]. Pura et al. also confirmed the tissue ingrowth into porous Ti implants (with a volume porosity of 65% and a mean pore size of 400 μm), releasing bisphosphonate alendronate [87]. In a more recent report by Markel et al., the authors showed that tenocytes favored small pores of 400 μm with a reduction of cell viability when samples with larger pores (700 and 1000 μm) were employed [88]. Conversely, the same group showed that fibroblasts preferred large pores [88]. Presumably, subsequent coating of the porous implant (physical, chemical and biological) would allow the soft-tissue seal to be enhanced and skin ingrowth around percutaneous implants [89]. Accordingly, in a previous study by our group, we showed that the fabrication of TiO_2_ nanotubes on the porous titanium facilitated fibroblast attachment in vitro and skin ingrowth in vivo in a rabbit model [20]. Subsequent formation of TiO_2_ nanolayer/Ag nanoparticle structures employing the atomic layer deposition (ALD) method not only facilitated the attachment of human fetal mesenchymal stem cells (FetMSCs) but also provided anti-bacterial effect against *Staphylococcus aureus* [90].

Apart from modification of the physical characteristics of the Ti implants in order to increase their biocompatibility and tissue regeneration, the implants can also be coated with either inorganic molecules (e.g., calcium phosphate (CaP), hydroxyapatite (HA), etc.) [91,92,93] or with various bioactive molecules (e.g., bone morphogenic protein, type I collagen, etc.) [94]. Indeed, several preclinical studies demonstrated the efficiency of the application of BMPs (including BMP-2 and BMP-7 derivates) to enhance osteogenesis and chondroblast activity, thus significantly improving osseointegration [95,96,97]. For example, Kim et al. showed that BMP-2 immobilized to the Ti implants’ surface significantly increased bone regeneration in dogs (implants were installed into maxilla and mandible bones) as compared to control implants without BMP-2 protein [98]. Presumably, these modifications could also be employed in future studies for our 3D-printed Ti implants.

## 5. Conclusions

Sintered Ti samples did not show significant toxicities towards MC3T3-E1 cells and dermal fibroblasts after 14 days of co-incubation (MTT assay) and were comparable to 3D-printed samples S1–S9. When osteogenic markers were evaluated, the 3D-printed S2–S6 samples were significantly better than the sintered Ti samples. When the adhesion markers and extracellular matrix components (fibronectin, vitronectin, type I collagen) of dermal fibroblasts were evaluated, the S6–S9 3D-printed samples were comparable to the sintered Ti samples, although some of the evaluated markers were high when 3D-printed discs were applied. In conclusion, 3D-printed Ti samples S6–S8 showed the best results for dermal fibroblast cells, and samples S3–S5 showed the best results for MC3T3 osteoblast cells. Considering that the possibility of 3D printing makes it possible to implement several optimal parameters for various cells and tissues (skin, bone tissue) in one implant, preference should be given to this approach. The data obtained provide a practical justification for the use of these Ti pore parameters to produce a transcutaneous osseointegrated implant design for preclinical in vivo studies and application of 3D printing technology in regenerative medicine.

## Figures and Tables

**Figure 1 nanomaterials-14-01484-f001:**
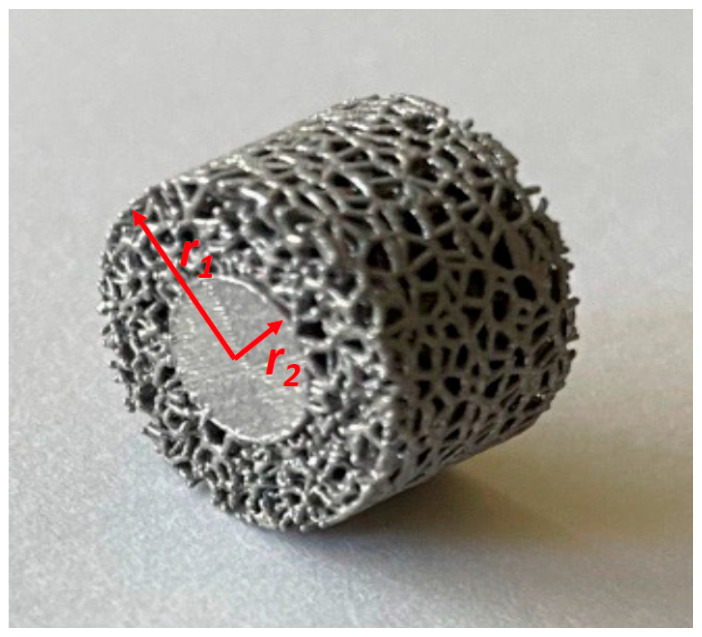
Example of 3D printed samples for the current study.

**Figure 2 nanomaterials-14-01484-f002:**
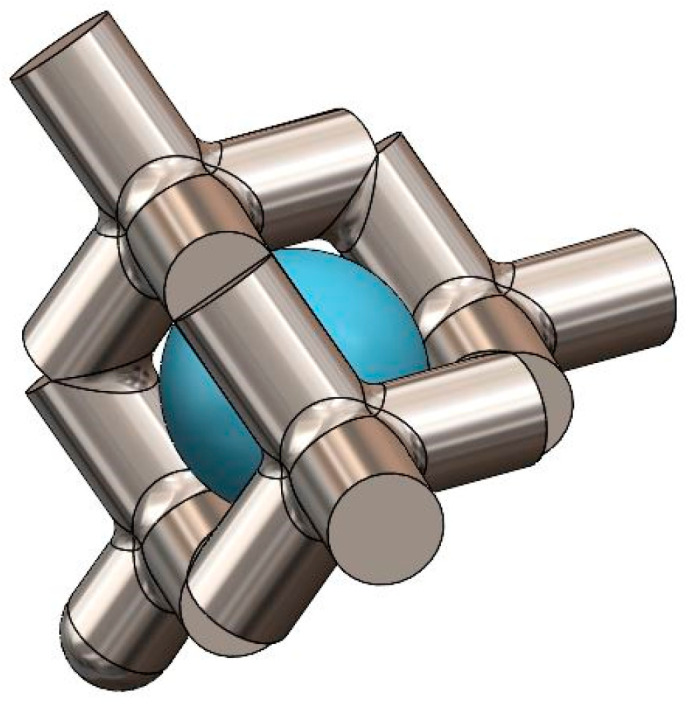
Lattice pore size example using a standard diamond cubic lattice structure. It should be noted that the ingrowth tabs utilized a Voronoi lattice structure, which is a randomized structure that results in varying pore diameters, which is why an average pore diameter was used.

**Figure 3 nanomaterials-14-01484-f003:**
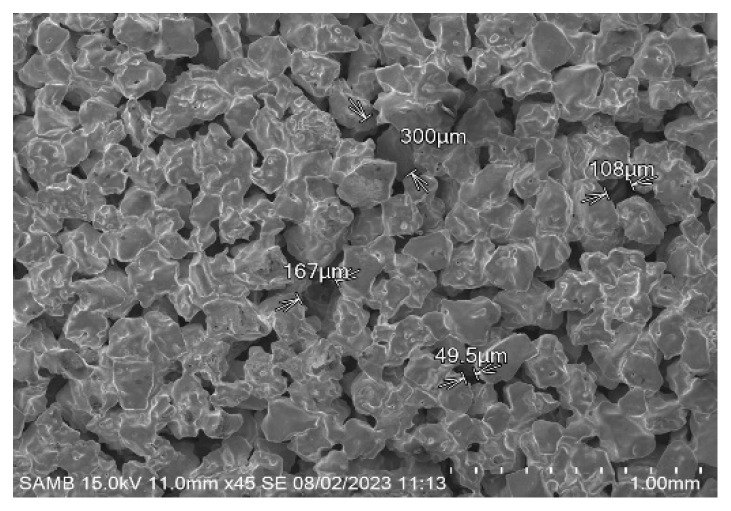
Structure of the sintered (control) samples.

**Figure 4 nanomaterials-14-01484-f004:**
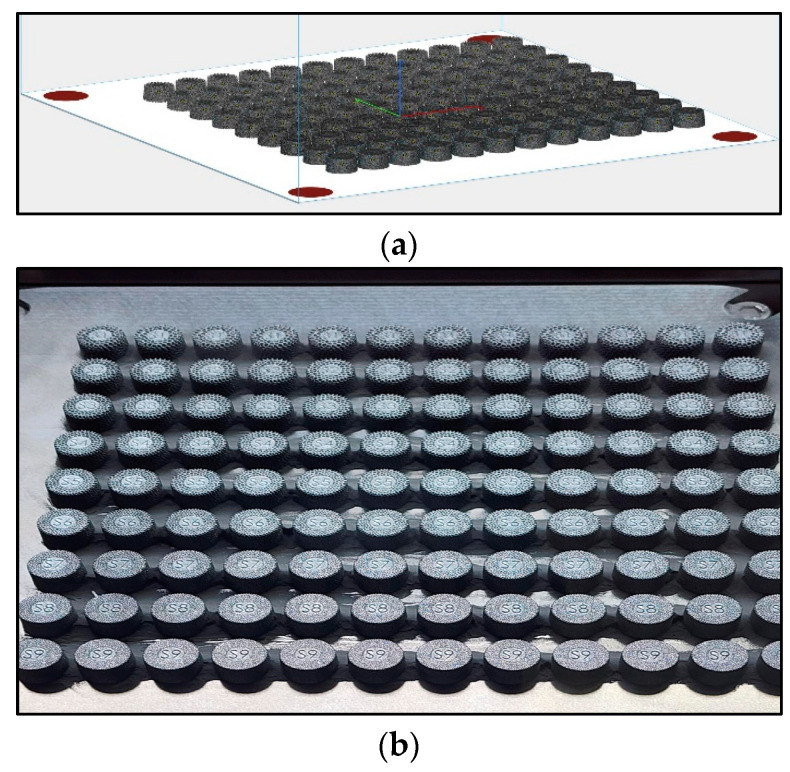
(**a**)—computer models of the tablets; (**b**)—3D printed tablets on the build plate.

**Figure 5 nanomaterials-14-01484-f005:**
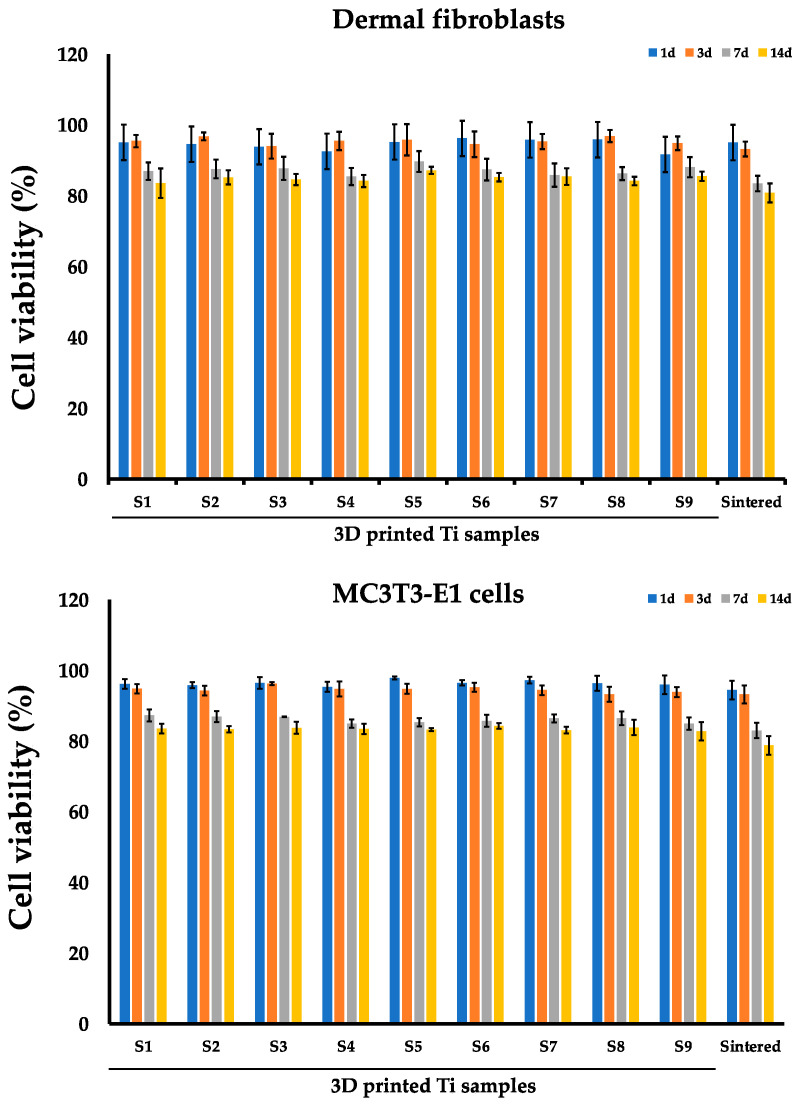
MTT assay of dermal fibroblasts and MC3T3-E1 osteoblast cells on sintered and 3D printed Ti samples S1–S9. Cell viability (%) was evaluated on the 1st, 3rd, 7th, and 14th day after co-incubation.

**Figure 6 nanomaterials-14-01484-f006:**
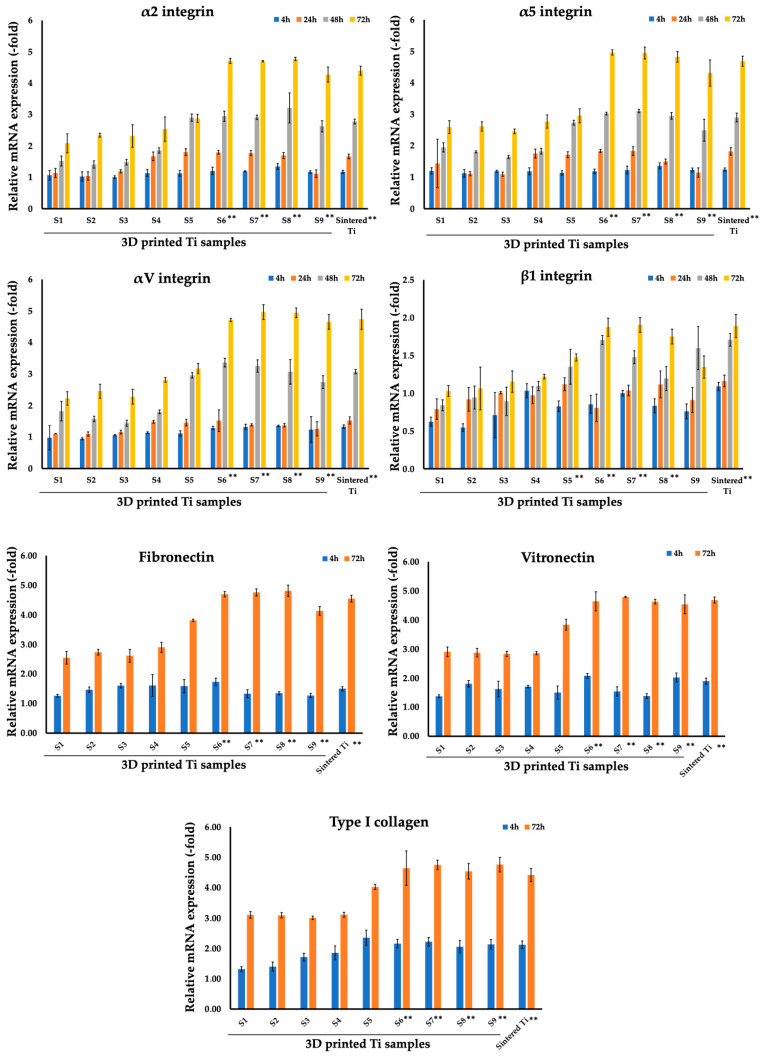
Comparison of expression of genes related to integrins and extracellular matrix components (fibronectin, vitronectin, type I collagen) of dermal fibroblasts on 3D-printed Ti samples S1–S9 and sintered Ti samples after 4, 24, 48, and 72 h of co-incubation. ** *p* < 0.01 for testing mean expression levels.

**Figure 7 nanomaterials-14-01484-f007:**
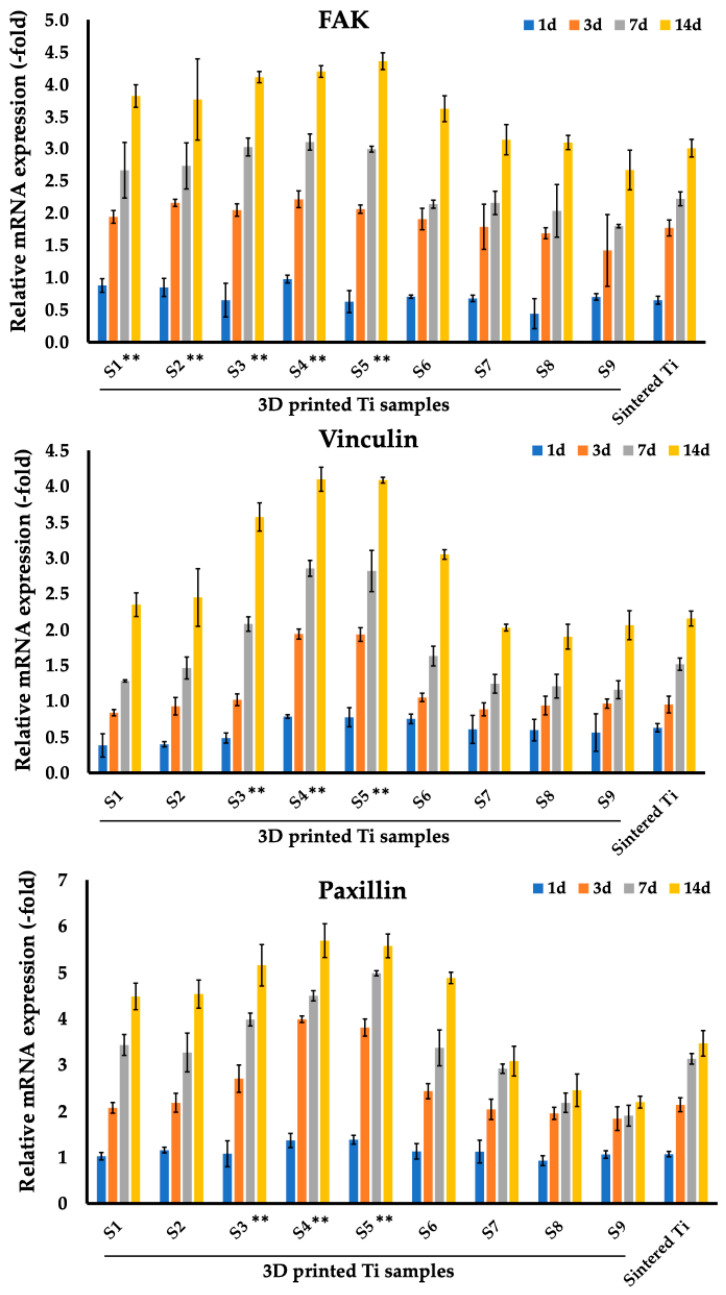
Comparison of expression of genes related to focal adhesion (FAK, vinculin, paxillin) markers of MC3T3-E1 cells between sintered Ti samples after 1, 3, 7, and 14 days of co-incubation and 3D printed T samples (S1–S9). ** *p* < 0.01 for testing mean expression levels.

**Figure 8 nanomaterials-14-01484-f008:**
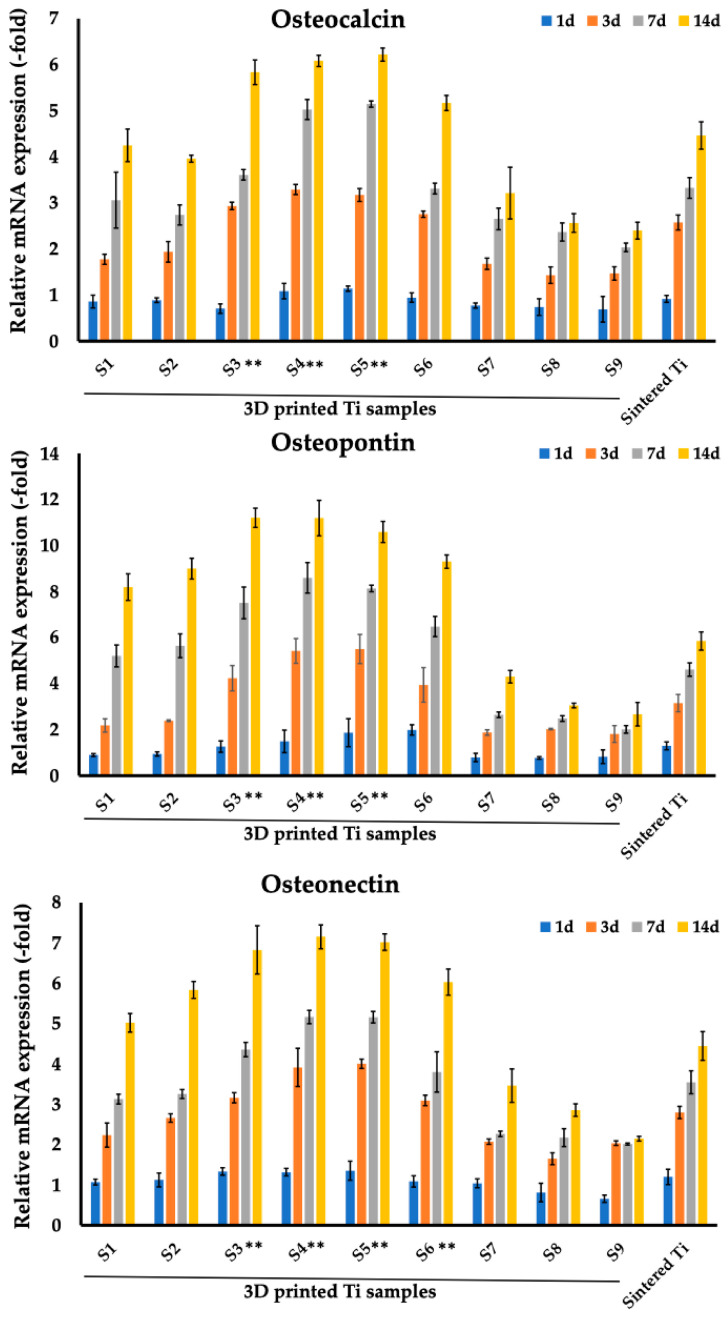
Comparison of expression of genes related to osteoblast-specific markers (osteocalcin, osteopontin, osteocalcin) of MC3T3-E1 cells between sintered Ti samples after 1, 3, 7, and 14 days of co-incubation and 3D-printed T samples (S1–S9). ** *p* < 0.01 for testing mean expression levels.

**Figure 9 nanomaterials-14-01484-f009:**
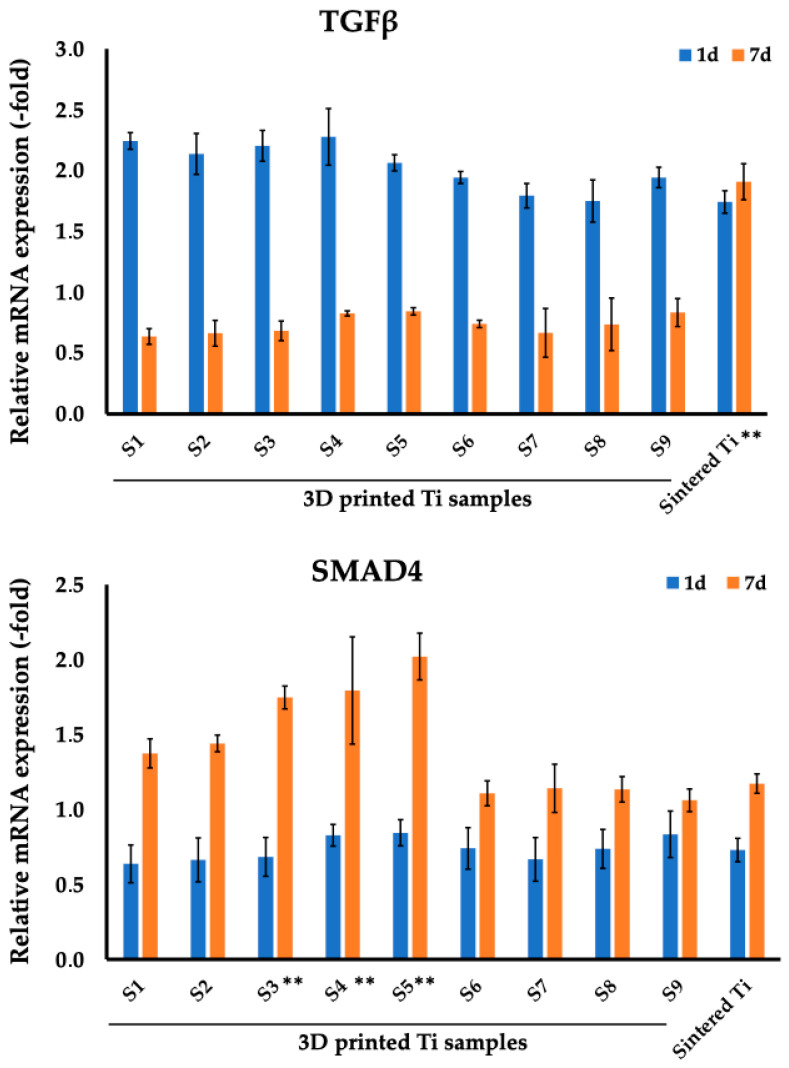
Comparison of gene expression profiles of TGF-β1 and SMAD4 in MC3T3-E1 osteoblast cells on days 1 and 7 after co-incubation between 3D printed and sintered Ti samples. ** *p* < 0.01 for testing mean expression levels.

**Table 2 nanomaterials-14-01484-t002:** Forward and reverse primers specific for tested genes used for RT-PCR studies.

Gene	Primers (5′-3′)	Product Length (bp)
Fibronectin	Fwd TGCAGTGGCTGAAGTCGCAAGG Rev GGGCTCCCCGTTTGAATTGCCA	119
Vitronectin	Fwd TGTTGATGCAGCGTTCGCCCT Rev TCCTGGCTGGGTTGCTGCTGAA	114
Type I collagen	Fwd CTCCTGACGCATGGCCAAGAA Rev TCAAGCATACCTCGGGTTTCCA	100
α2 integrin	Fwd AAGTGCCCTGTGGACCTACCCA Rev TGGTGAGGGTCAATCCCAGGCT	119
α5 integrin	Fwd ACCACCTGCAGAAACGAGAGGC Rev TGGCCCAAACTCACAGCGCA	111
αV integrin	Fwd TCCCACCGCAGGCTGACTTCAT Rev TCGGGTTTCCAAGGTCGCACAC	121
β1 integrin	Fwd TTCAGACTTCCGCATTGGCT Rev AATGGGCTGGTGCAGTTTTG	122
FAK	Fwd AGCACCTGGCCACCTAAGCAAC Rev CATTGGACCGGTCAAGGTTGGCA	125
Paxillin	Fwd AGGGCCTGGAACAGAGAGTGGA Rev AGCTGCTCCCAGTTTTCCCCTG	129
Vinculin	Fwd TCAAGCTGTTGGCAGTAGCCGC Rev TCTCTGCTGTGGCTCCAAGCCT	120
Osteocalcin	Fwd AGCAGGAGGGCAATAAGGTAGT Rev TCGTCACAAGCAGGGTTAAGC	118
Osteopontin	Fwd TGATTCTGGCAGCTCAGAGGA Rev CATTCTGTGGCGCAAGGAGATT	110
Osteonectin	Fwd ATGTCCTGGTCACCTTGTACGA Rev TCCAGGCGCTTCTCATTCTCAT	103
TGF-β1	Fwd ACCCGCGTGCTAATGGTGGA Rev GGGCACTGCTTCCCGAATGTCT	111
SMAD4	Fwd AGCCAGGACAGCAGCAGAATGGA Rev ATGGCCGTTTTGGTGGTGAGGC	128

**Table 3 nanomaterials-14-01484-t003:** Mean (with standard deviation) cell viability (%) of dermal fibroblasts on 3D-printed S1–S9 and on the sintered Ti sample after 1, 3, 7, and 14 days of co-incubation.

	S1	S2	S3	S4	S5	S6	S7	S8	S9	Sintered Ti
Day 1	95.10 (1.05)	94.57 (2.29)	93.83 (1.90)	92.53 (2.20)	95.20 (3.87)	96.20 (1.11)	95.80 (4.24)	95.83 (1.48)	91.67 (1.97)	95.02 (2.14)
Day 3	95.43 (1.76)	96.77 (1.10)	94.00 (3.48)	95.47 (2.58)	95.80 (4.42)	94.53 (3.65)	95.30 (2.15)	96.87 (1.71)	94.83 (1.91)	93.20 (2.08)
Day 7	86.93 (2.48)	87.57 (2.63)	87.73 (3.29)	85.43 (2.43)	89.67 (2.97)	87.40 (3.04)	85.83 (3.31)	86.27 (1.83)	88.07 (2.84)	83.46 (2.19)
Day 14	83.57 (4.15)	85.20 (2.00)	84.60 (1.57)	84.17 (1.71)	87.20 (1.01)	85.27 (1.18)	85.43 (2.35)	84.17 (1.22)	85.50 (1.30)	80.79 (2.69)

**Table 4 nanomaterials-14-01484-t004:** Mean (with standard deviation) cell viability (%) of MC3T3-E1 osteoblast cells on 3D-printed S1–S9 and on the sintered Ti sample after 1, 3, 7 and 14 days of co-incubation.

	S1	S2	S3	S4	S5	S6	S7	S8	S9	Sintered Ti
Day 1	96.20 (1.35)	95.83 (0.87)	96.47 (1.66)	95.37 (1.46)	97.87 (0.45)	96.50 (0.79)	97.27 (0.95)	96.37 (2.14)	95.93 (2.59)	94.41 (2.62)
Day 3	94.80 (1.31)	94.27 (1.35)	96.27 (0.45)	94.77 (2.14)	94.77 (1.42)	95.20 (1.30)	94.40 (1.37)	93.27 (2.10)	93.87 (1.42)	93.23 (2.55)
Day 7	87.30 (1.71)	86.93 (1.58)	86.90 (0.10)	84.93 (1.21)	85.33 (1.18)	85.77 (1.71)	86.43 (1.12)	86.47 (1.96)	84.93 (1.76)	83.02 (2.18)
Day 14	83.53 (0.78)	83.33 (1.59)	83.77 (1.69)	83.47 (1.63)	83.23 (2.30)	84.30 (1.31)	83.10 (0.98)	83.87 (1.53)	82.77 (1.42)	78.90 (2.98)

**Table 5 nanomaterials-14-01484-t005:** Mean (with standard deviation) of expression of genes related to integrins and extracellular matrix components (fibronectin, vitronectin, type I collagen) of dermal fibroblasts on 3D-printed Ti samples S1–S9 and sintered Ti samples after 4, 24, 48 and 72 h of co-incubation.

**α2 Integrin**										
	**S1**	**S2**	**S3**	**S4**	**S5**	**S6**	**S7**	**S8**	**S9**	**Sintered Ti**
4 h	1.07 (0.15)	1.03 (0.15)	1.01 (0.04)	1.14 (0.11)	1.13 (0.09)	1.21 (0.12)	1.19 (0.02)	1.35 (0.09)	1.17 (0.04)	1.18 (0.05)
24 h	1.14 (0.14)	1.04 (0.13)	1.19 (0.05)	1.67 (0.14)	1.80 (0.11)	1.80 (0.06)	1.77 (0.08)	1.70 (0.09)	1.12 (0.12)	1.67 (0.08)
48 h	1.52 (0.16)	1.41 (0.12)	1.48 (0.09)	1.86 (0.08)	2.90 (0.12)	2.95 (0.16)	2.91 (0.07)	3.21 (0.48)	2.62 (0.18)	2.78 (0.07)
72 h	2.08 (0.30)	2.34 (0.06)	2.32 (0.36)	2.53 (0.39)	2.88 (0.13)	4.71 (0.08)	4.70 (0.02)	4.77 (0.06)	4.27 (0.24)	4.40 (0.14)
**α5 Integrin**										
	**S1**	**S2**	**S3**	**S4**	**S5**	**S6**	**S7**	**S8**	**S9**	**Sintered Ti**
4 h	1.20 (0.10)	1.13 (0.12)	1.19 (0.03)	1.19 (0.11)	1.14 (0.07)	1.19 (0.07)	1.23 (0.12)	1.36 (0.09)	1.23 (0.06)	1.25 (0.05)
24 h	1.44 (0.77)	1.11 (0.07)	1.10 (0.07)	1.75 (0.14)	1.72 (0.09)	1.84 (0.05)	1.83 (0.14)	1.50 (0.08)	1.15 (0.15)	1.82 (0.12)
48 h	1.95 (0.15)	1.81 (0.03)	1.65 (0.05)	1.83 (0.09)	2.73 (0.08)	3.02 (0.04)	3.10 (0.06)	2.95 (0.11)	2.49 (0.35)	2.90 (0.14)
72 h	2.59 (0.20)	2.61 (0.15)	2.46 (0.08)	2.77 (0.21)	2.95 (0.22)	4.96 (0.08)	4.95 (0.19)	4.82 (0.17)	4.31 (0.42)	4.69 (0.16)
**αV Integrin**										
	**S1**	**S2**	**S3**	**S4**	**S5**	**S6**	**S7**	**S8**	**S9**	**Sintered Ti**
4 h	0.98 (0.38)	0.94 (0.03)	1.06 (0.02)	1.14 (0.03)	1.11 (0.09)	1.29 (0.06)	1.32 (0.08)	1.36 (0.02)	1.24 (0.41)	1.33 (0.06)
24 h	1.10 (0.01)	1.10 (0.07)	1.16 (0.06)	1.48 (0.05)	1.46 (0.10)	1.52 (0.34)	1.39 (0.04)	1.37 (0.06)	1.26 (0.23)	1.53 (0.11)
48 h	1.82 (0.31)	1.57 (0.09)	1.44 (0.10)	1.80 (0.06)	2.96 (0.09)	3.36 (0.14)	3.25 (0.20)	3.07 (0.39)	2.74 (0.20)	3.07 (0.07)
72 h	2.22 (0.21)	2.45 (0.22)	2.28 (0.24)	2.81 (0.07)	3.17 (0.16)	4.71 (0.05)	4.96 (0.23)	4.94 (0.15)	4.65 (0.23)	4.73 (0.32)
**β1 Integrin**										
	**S1**	**S2**	**S3**	**S4**	**S5**	**S6**	**S7**	**S8**	**S9**	**Sintered Ti**
4 h	0.62 (0.06)	0.55 (0.05)	0.71 (0.30)	1.03 (0.09)	0.83 (0.07)	0.85 (0.12)	1.00 (0.04)	0.83 (0.09)	0.76 (0.10)	1.09 (0.05)
24 h	0.79 (0.14)	0.92 (0.16)	1.01 (0.02)	0.97 (0.11)	1.12 (0.09)	0.81 (0.18)	1.04 (0.07)	1.12 (0.18)	0.91 (0.17)	1.16 (0.08)
48 h	0.84 (0.07)	0.94 (0.15)	0.89 (0.19)	1.10 (0.06)	1.35 (0.23)	1.70 (0.06)	1.48 (0.09)	1.19 (0.16)	1.60 (0.29)	1.71 (0.08)
72 h	1.03 (0.07)	1.06 (0.28)	1.15 (0.14)	1.22 (0.03)	1.47 (0.05)	1.87 (0.12)	1.90 (0.10)	1.75 (0.10)	1.35 (0.15)	1.89 (0.15)
**Fibronectin**										
	**S1**	**S2**	**S3**	**S4**	**S5**	**S6**	**S7**	**S8**	**S9**	**Sintered Ti**
4 h	1.26 (0.05)	1.47 (0.09)	1.61 (0.07)	1.61 (0.37)	1.59 (0.22)	1.73 (0.13)	1.33 (0.14)	1.35 (0.05)	1.27 (0.07)	1.50 (0.07)
72 h	2.55 (0.22)	2.74 (0.10)	2.61 (0.22)	2.90 (0.17)	3.81 (0.04)	4.70 (0.09)	4.76 (0.12)	4.81 (0.19)	4.13 (0.15)	4.55 (0.11)
**Vitronectin**										
	**S1**	**S2**	**S3**	**S4**	**S5**	**S6**	**S7**	**S8**	**S9**	**Sintered Ti**
4 h	1.38 (0.05)	1.81 (0.11)	1.62 (0.27)	1.71 (0.04)	1.50 (0.23)	2.08 (0.08)	1.54 (0.16)	1.38 (0.08)	2.02 (0.15)	1.89 (0.10)
72 h	2.90 (0.17)	2.87 (0.15)	2.83 (0.09)	2.86 (0.06)	3.84 (0.19)	4.64 (0.33)	4.79 (0.02)	4.63 (0.08)	4.54 (0.33)	4.69 (0.10)
**Type I Collagen**										
	**S1**	**S2**	**S3**	**S4**	**S5**	**S6**	**S7**	**S8**	**S9**	**Sintered Ti**
4 h	1.32 (0.08)	1.40 (0.15)	1.71 (0.13)	1.85 (0.23)	2.35 (0.25)	2.16 (0.14)	2.22 (0.14)	2.05 (0.20)	2.13 (0.16)	2.12 (0.12)
72 h	3.10 (0.11)	3.09 (0.09)	3.01 (0.06)	3.11 (0.08)	4.02 (0.09)	4.64 (0.57)	4.75 (0.16)	4.54 (0.26)	4.76 (0.24)	4.42 (0.22)

**Table 6 nanomaterials-14-01484-t006:** Mean (with standard deviation) gene expression related to focal adhesion (FAK, vinculin, paxillin) of MC3T3-E1 cells on 3D printed Ti samples S1–S9 and on the sintered Ti sample after 1, 3, 7, and 14 days of co-incubation.

	FAK				Vincullin				Paxillin			
Samples	Day 1	Day 3	Day 7	Day 14	Day 1	Day 3	Day 7	Day 14	Day 1	Day 3	Day 7	Day 14
S1	0.88 (0.11)	1.94 (0.1)	2.67 (0.43)	3.82 (0.17)	0.38 (0.16)	0.84 (0.04)	1.28 (0.02)	2.35 (0.17)	1.02 (0.08)	2.07 (0.12)	3.43 (0.23)	4.48 (0.29)
S2	0.85 (0.14)	2.16 (0.06)	2.73 (0.36)	3.76 (0.63)	0.4 (0.04)	0.93 (0.12)	1.46 (0.15)	2.45 (0.4)	1.16 (0.06)	2.18 (0.2)	3.27 (0.42)	4.53 (0.3)
S3	0.65 (0.26)	2.05 (0.1)	3.03 (0.14)	4.11 (0.09)	0.49 (0.07)	1.02 (0.08)	2.08 (0.1)	3.57 (0.2)	1.08 (0.28)	2.7 (0.3)	3.98 (0.14)	5.15 (0.45)
S4	0.98 (0.06)	2.22 (0.13)	3.1 (0.13)	4.2 (0.09)	0.79 (0.03)	1.94 (0.07)	2.85 (0.11)	4.1 (0.17)	1.36 (0.15)	3.99 (0.07)	4.49 (0.11)	5.69 (0.37)
S5	0.63 (0.17)	2.06 (0.06)	2.99 (0.05)	4.36 (0.13)	0.78 (0.13)	1.93 (0.1)	2.82 (0.29)	4.08 (0.04)	1.38 (0.1)	3.81 (0.19)	4.98 (0.06)	5.57 (0.26)
S6	0.71 (0.03)	1.91 (0.17)	2.14 (0.06)	3.62 (0.2)	0.75 (0.07)	1.05 (0.06)	1.63 (0.14)	3.05 (0.07)	1.13 (0.17)	2.43 (0.16)	3.37 (0.39)	4.88 (0.12)
S7	0.68 (0.05)	1.79 (0.35)	2.16 (0.18)	3.14 (0.23)	0.61 (0.2)	0.89 (0.09)	1.24 (0.13)	2.03 (0.05)	1.12 (0.25)	2.04 (0.22)	2.92 (0.1)	3.08 (0.32)
S8	0.44 (0.23)	1.69 (0.09)	2.04 (0.41)	3.1 (0.11)	0.6 (0.15)	0.94 (0.13)	1.21 (0.17)	1.9 (0.17)	0.93 (0.11)	1.95 (0.13)	2.18 (0.21)	2.45 (0.35)
S9	0.7 (0.05)	1.42 (0.56)	1.8 (0.03)	2.67 (0.31)	0.56 (0.26)	0.97 (0.06)	1.16 (0.13)	2.06 (0.2)	1.06 (0.08)	1.84 (0.26)	1.9 (0.23)	2.19 (0.13)
Sintered Ti	0.65 (0.06)	1.77 (0.12)	2.22 (0.11)	3.01 (0.14)	0.63 (0.06)	0.95 (0.12)	1.52 (0.09)	2.15 (0.1)	1.07 (0.06)	2.14 (0.15)	3.13 (0.12)	3.47 (0.28)

**Table 7 nanomaterials-14-01484-t007:** Mean (with standard deviation) gene expression related to osteoblast-specific markers (osteocalcin, osteopontin, osteocalcin) of MC3T3-E1 cells on 3D printed Ti samples S1–S9 and on the sintered Ti sample after 1, 3, 7 and 14 days of co-incubation.

	Osteocalcin				Osteopontin				Osteonectin			
Samples	Day 1	Day 3	Day 7	Day 14	Day 1	Day 3	Day 7	Day 14	Day 1	Day 3	Day 7	Day 14
S1	0.86 (0.14)	1.78 (0.11)	3.06 (0.61)	4.25 (0.35)	0.9 (0.06)	2.18 (0.29)	5.2 (0.47)	8.19 (0.58)	1.07 (0.07)	2.24 (0.3)	3.13 (0.12)	5.02 (0.23)
S2	0.89 (0.05)	1.94 (0.22)	2.74 (0.22)	3.96 (0.08)	0.94 (0.09)	2.39 (0.04)	5.64 (0.52)	8.99 (0.45)	1.12 (0.17)	2.66 (0.1)	3.26 (0.11)	5.84 (0.21)
S3	0.71 (0.1)	2.94 (0.08)	3.61 (0.11)	5.83 (0.27)	1.26 (0.25)	4.23 (0.55)	7.5 (0.69)	11.2 (0.42)	1.33 (0.09)	3.16 (0.13)	4.36 (0.18)	6.83 (0.6)
S4	1.09 (0.17)	3.29 (0.11)	5.02 (0.22)	6.08 (0.12)	1.49 (0.49)	5.42 (0.54)	8.59 (0.66)	11.19 (0.77)	1.32 (0.09)	3.92 (0.48)	5.17 (0.17)	7.15 (0.29)
S5	1.14 (0.06)	3.17 (0.14)	5.14 (0.07)	6.22 (0.14)	1.87 (0.61)	5.5 (0.63)	8.13 (0.14)	10.59 (0.46)	1.35 (0.24)	4.01 (0.12)	5.16 (0.14)	7.02 (0.2)
S6	0.95 (0.1)	2.76 (0.07)	3.31 (0.12)	5.17 (0.16)	1.99 (0.22)	3.94 (0.75)	6.48 (0.44)	9.3 (0.29)	1.09 (0.14)	3.1 (0.13)	3.8 (0.5)	6.03 (0.32)
S7	0.77 (0.06)	1.68 (0.12)	2.65 (0.23)	3.21 (0.56)	0.79 (0.18)	1.87 (0.12)	2.64 (0.12)	4.3 (0.27)	1.04 (0.11)	2.07 (0.07)	2.27 (0.07)	3.46 (0.41)
S8	0.74 (0.18)	1.43 (0.18)	2.37 (0.2)	2.57 (0.2)	0.77 (0.05)	2.02 (0.03)	2.48 (0.12)	3.05 (0.1)	0.81 (0.23)	1.65 (0.15)	2.17 (0.22)	2.86 (0.16)
S9	0.69 (0.28)	1.47 (0.15)	2.04 (0.09)	2.4 (0.18)	0.82 (0.29)	1.81 (0.36)	2.01 (0.17)	2.67 (0.51)	0.66 (0.09)	2.04 (0.06)	2.01 (0.03)	2.15 (0.06)
Sintered Ti	0.92 (0.08)	2.58 (0.16)	3.33 (0.22)	4.46 (0.3)	1.3 (0.17)	3.15 (0.38)	4.61 (0.29)	5.85 (0.4)	1.2 (0.19)	2.8 (0.15)	3.55 (0.29)	4.45 (0.36)

**Table 8 nanomaterials-14-01484-t008:** Mean (with standard deviation) gene expression profiles of TGF-β1 and SMAD4 in MC3T3-E1 osteoblast cells on days 1 and 7 after co-incubation between 3D printed and sintered Ti samples.

**TGF-β1**										
	**S1**	**S2**	**S3**	**S4**	**S5**	**S6**	**S7**	**S8**	**S9**	**Sintered Ti**
Day 1	2.24 (0.07)	2.14 (0.17)	2.20 (0.13)	2.28 (0.23)	2.06 (0.07)	1.94 (0.05)	1.79 (0.10)	1.75 (0.17)	1.94 (0.08)	1.74 (0.09)
Day 7	0.64 (0.06)	0.66 (0.11)	0.68 (0.08)	0.83 (0.02)	0.84 (0.03)	0.74 (0.03)	0.67 (0.20)	0.74 (0.22)	0.83 (0.12)	1.91 (0.15)
**SMAD4**										
	**S1**	**S2**	**S3**	**S4**	**S5**	**S6**	**S7**	**S8**	**S9**	**Sintered Ti**
Day 1	0.64 (0.13)	0.66 (0.15)	0.68 (0.13)	0.83 (0.07)	0.84 (0.09)	0.74 (0.14)	0.67 (0.14)	0.74 (0.13)	0.83 (0.16)	0.73 (0.09)
Day 7	1.37 (0.10)	1.44 (0.06)	1.75 (0.07)	1.79 (0.36)	2.02 (0.16)	1.11 (0.08)	1.14 (0.16)	1.13 (0.09)	1.06 (0.08)	1.17 (0.06)

## Data Availability

The datasets used and/or analyzed during the current study are available from the corresponding authors, Maxim Shevtsov and Mark Pitkin, on reasonable request.

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
