# Peer review of "Comparison In Vitro Study on the Interface between Skin and Bone Cell Cultures and Microporous Titanium Samples Manufactured with 3D Printing Technology Versus Sintered Samples"

_nanomaterials, 2024, doi:10.3390/nano14181484_

Round 1

Reviewer 1 Report

Comments and Suggestions for Authors

attached

Comments on the Quality of English Language

None

Author Response

We would like to thank the reviewer for the provided comments. Please find the responses to the comments below.

COMMENT 1: It should be more convincing if the authors could add the blank control group in MTT assay in Figure 4.

ANSWER 1: We have added the description of the blank control cells into the results section 3.1 as follows: In the blank control group when cells were cultured in cultural flasks without sintered or 3D printed implants the cell viability (%) for dermal fibroblasts on the 1st, 3rd, 7th, and 14th days constituted 97.41 ± 1.18%, 94.32 ± 2.21%, 86.78 ± 4.61%, and 85.53 ± 4.19%, respectively. The cell viability (%) in the blank control group for MC3T3-E1 osteoblasts on the 1st, 3rd, 7th, and 14th days constituted 95.37 ± 3.21%, 96.52 ± 1.89%, 89.91 ± 4.81%, and 84.92 ± 3.73%, respectively.

COMMENT 2: The figures would be more elicit if the authors could use “*” to label the statistical

significance among the S1-9 groups in Figure 5 and Figure 6.

ANSWER 2: We have added this information into the figures.

COMMENT 3: The authors need to make the name uniform in the manuscript. For example, the authors wrote “3D printed” in the abstract section and “e-printed” in the results section.

ANSWER 3: We have unified the terminology throughout the manuscript using “3D printed”.

COMMENT 4:  Please interpret why the authors chose osteocalcin, osteopontin, osteocalcin as osteoblast specific markers at 1, 3, 7, 14 days of coincubation and then do the rt-PCR of TGF-β1 and SMAD4 at day 1 and 7 again?

ANSWER 4: In the current study we assessed specific genes involved in osteogenesis - osteopontin, osteonectin and osteocalcin. Additionally, we evaluated TGF-β1 and SMAD4 involved in the signaling pathway regulating this process as we wanted also to confirm that particularly this pathway is involved in the osteogenesis of cells co-incubated on our sintered and 3D printed Ti samples. TGF-β1 (via canonical and non-canonical pathways) plays a key role in osteoblast growth and differentiation and regulation of osteoclastogenesis, and SMAD4 regulates osteoblast and osteocyte viability.

Reviewer 2 Report

Comments and Suggestions for Authors

This is a very elegant and insightful study regarding the in vitro study on the interface between skin and bone cell cultures and nanoporous titanium samples of Skin and Bone Integrated Pylons (SBIP) manufactured with 3-D printing technology versus sintered samples. Indeed, this subject is interesting due to the lack of alternatives in such percutaneous materials.

However, some changes should be made before considering the manuscript for publication:

1)      Title: In-vitro should be “In vitro”.

2)      Page 2, Lines 68: The sentence “For example, an otherwise effective published study in sheep showed a reduction in skin infection rate to 16.7% over a 24-month period.” Should be rephrased. Authors should change the “For example” and the “otherwise”.

3)     Page 2, Lines 72: The sentence “This negative result may be explained in part that as in some other experiments with porous implants [7-9], the porous cladding was relatively thin.” Should be rephrased once it does not mean anything.

4)    Page 2, Line 79: The sentence ““volume fraction,” with is the” should be rewritten.

5)    Page 2, Line 83: The sentence “The previous SBIP samples for animal studies were fabricated with sintering of titanium particles in boron nitride molds. That…” should be rewritten once the reader does not know what previous mean.

6)    Page 2, Line 94: The authors used “we employed”. The authors use the first person plural throughout most of the entire document. They should replace it throughout the document with the third person plural or 'The authors'.

7) Tables 1 and 2 present the same results as Figure 4, and Table 3 presents the same results as Figure 5. The results should be presented either as a table or a figure, not both.

The results are very interesting and the discussion and conclusion are supported by the results. Therefore, in my opinion, after these changes the manuscript is publishable.

Author Response

We would like to thank the reviewer for the provided comments. Please find the responses to the comments below.

COMMENT 1: Title: In-vitro should be “In vitro”.

ANSWER 1: We have corrected this.

COMMENT 2: Page 2, Lines 68: The sentence “For example, an otherwise effective published study in sheep showed a reduction in skin infection rate to 16.7% over a 24-month period.” Should be rephrased. Authors should change the “For example” and the “otherwise”.

ANSWER 2: We have rephrased this sentence.

COMMENT 3: Page 2, Lines 72: The sentence “This negative result may be explained in part that as in some other experiments with porous implants [7-9], the porous cladding was relatively thin.” Should be rephrased once it does not mean anything.

ANSWER 3: We have rephrased this sentence as follows: This negative result may be explained in part by the fact that the researchers either did not use porous titanium at all in the implant design or employed relatively thin porous cladding, which in both cases did not lead to the desired biointegration of the implant with the surrounding tissues.

COMMENT 4: Page 2, Line 79: The sentence ““volume fraction,” with is the” should be rewritten.

ANSWER 4: We have corrected this sentence.

COMMENT 5: Page 2, Line 83: The sentence “The previous SBIP samples for animal studies were fabricated with sintering of titanium particles in boron nitride molds. That…” should be rewritten once the reader does not know what previous mean.

ANSWER 5: We have corrected this sentence.

COMMENT 6: Page 2, Line 94: The authors used “we employed”. The authors use the first person plural throughout most of the entire document. They should replace it throughout the document with the third person plural or 'The authors'.

ANSWER 6: This was corrected.

COMMENT 7: Tables 1 and 2 present the same results as Figure 4, and Table 3 presents the same results as Figure 5. The results should be presented either as a table or a figure, not both.

ANSWER 7: We felt that the tables provide additional specific information for readers and interested researchers that cannot be directly obtained from the figures presented, in particular the expression values ​​of various markers, which researchers can then apply to their own studies.

COMMENT 8: The results are very interesting and the discussion and conclusion are supported by the results. Therefore, in my opinion, after these changes the manuscript is publishable.

ANSWER 8: We would like to thank the reviewer for his good evaluation of our work.

Reviewer 3 Report

Comments and Suggestions for Authors

The current study presents an interesting comparison of various in vitro interfaces between skin and bone cells and nanoporous titanium scaffolds; however, there are some issues that require consideration before accepting this article for publication. The following major concerns are as follows: 

The article title is very long; it needs to be more concise.

In the abstract, include the values of important results achieved.

In the introduction, it is too much to support one sentence with more than 25 references; I think up to 5 references is fine.

The article's introduction should clearly state the study's aim.

According to Line 112, the fabricated and tested Ti discs have nanoporoes, but Section 2.1 contradicts this by stating that the pore size ranges from 900 to 90 Mm.

Some times in the research article, the authors use Ti discs, and other times they use Ti scaffolds; this is also not consistent.

In the methodology section, the authors must include a table that contains the microstructure specifications of the nine sets, such as porosity (%), avarage pore size, pore wall thicknesses, and so on.

It is important to emphasise the differences between sintered and non-sinterred samples in the results and discussion sections.

The discussion did not include samples S1, S2, and S9, and it neglected to compare the sintered and non-sinterred samples in any way.

Author Response

We would like to thank the reviewer for the provided comments. Please find the responses to the comments below.

COMMENT 1: In the abstract, include the values of important results achieved.

ANSWER 1: We have corrected the abstract.

COMMENT 2: In the introduction, it is too much to support one sentence with more than 25 references; I think up to 5 references is fine.

ANSWER 2: We felt that the references provided would help interested researchers to become more familiar with the implant designs used in this study.

COMMENT 3: The article's introduction should clearly state the study's aim.

ANSWER 3: We have rephrased the introduction section.

COMMENT 4: According to Line 112, the fabricated and tested Ti discs have nanoporoes, but Section 2.1 contradicts this by stating that the pore size ranges from 900 to 90 Mm.

ANSWER 4: We have corrected the misprint in the section.

COMMENT 5; Some times in the research article, the authors use Ti discs, and other times they use Ti scaffolds; this is also not consistent.

ANSWER 5: We felt that both terms reflect the essence of the samples used. In essence, they are a scaffold for the integration of skin and bone tissue.

COMMENT 6: In the methodology section, the authors must include a table that contains the microstructure specifications of the nine sets, such as porosity (%), avarage pore size, pore wall thicknesses, and so on.

ANSWERS 6: We have added the descriptive Table 1 into the methodology section.

COMMENT 7: It is important to emphasise the differences between sintered and non-sinterred samples in the results and discussion sections. The discussion did not include samples S1, S2, and S9, and it neglected to compare the sintered and non-sinterred samples in any way.

ANSWER 7: We have added this information.

Round 2

Reviewer 1 Report

Comments and Suggestions for Authors

 Maybe the authors misunderstand the meaning of Q2. It could be apparent if the authors labeled the statistical significance between the different groups or within groups in the figures. For example, "*" represented p<0.05, and "***" represented p<0.001.

Comments on the Quality of English Language

1. If possible, please give a PDF version of the manuscript with highlighted content instead of a modified version in Word or WPS.

2. Please improve the quality of the English language. It is improper that the authors use so much passive tense in the manuscript.

Author Response

We would like to thank the reviewer for the provided comments. We have revised the manuscript accordingly. Please find the reply for the comments below.

COMMENT 1: Maybe the authors misunderstand the meaning of Q2. It could be apparent if the authors labeled the statistical significance between the different groups or within groups in the figures. For example, "*" represented p<0.05, and "***" represented p<0.001.

ANSWER 1: We thank the reviewer for the clarification. The degree of significance (below p<0.01 in all cases) is now explicit. Upon re-analyzing the data, one additional group of significantly high expressions (FAK, Figure 6) was found and correspondingly notated.

Comments on the Quality of English Language

COMMENT 2: If possible, please give a PDF version of the manuscript with highlighted content instead of a modified version in Word or WPS.

ANSWER 2: We tried to change corrections to the highlighted but the program do not provide this.

COMMENT 3: Please improve the quality of the English language. It is improper that the authors use so much passive tense in the manuscript.

ANSWER 3: We have double checked the manuscript for English readability and grammatical accuracy and corrected it.

Reviewer 2 Report

Comments and Suggestions for Authors

The manuscript has improved. I think there is a formatting problem with Figure 1 and order of the rest of the Figures. Once corrected/verified the manuscript is publishable.

Author Response

We would like to thank the reviewer for the provided comments. We have revised the manuscript accordingly. Please find the response below.

COMMENT 1: The manuscript has improved. I think there is a formatting problem with Figure 1 and order of the rest of the Figures. Once corrected/verified the manuscript is publishable.

ANSWER 1: We have edited manuscript in full for English readability and grammatical accuracy and checked the figures.

Reviewer 3 Report

Comments and Suggestions for Authors

Authors have covered most of the raised issues, and the manuscript in its current form is accepted for publication.

Author Response

COMMENT 1: Authors have covered most of the raised issues, and the manuscript in its current form is accepted for publication.

ANSWER 1: We thank the reviewer for his time and consideration to assess our manuscript.